Ambient ozone pollution at a coal chemical industry city in the border of Loess Plateau and Mu Us Desert: characteristics, sensitivity analysis and control strategies

Yin Manfei 1 2
Zhang Xin 2 3
Li Yunfeng 2 3
Fan Kai 4
Li Hong 2
Gao Rui gaorui@craes.org.cn 2
Li Jinjuan summy_lee@163.com 1
1 College of Resource and Environment Engineering, Guizhou University , Guiyang , China
2 State Key Laboratory of Environmental Criteria and Risk Assessment, Chinese Research Academy of Environmental Sciences , Beijing , China
3 Environment Research Institute, Shandong University , Jinan , China
4 Yulin Municipal Ecology and Environment Bureau , Yulin , China
Wang Xinfeng
Electronic publication date: 2021 Apr 27
Publication date: 2021
Volume: 9
Electronic Location ID: e11322
Received 2021 Jan 12; Accepted 2021 Mar 31
Copyright: ©2021 Yin et al.
Copyright year: 2021
Copyright holder: Yin et al.
License: This is an open access article distributed under the terms of the Creative Commons Attribution License, which permits unrestricted use, distribution, reproduction and adaptation in any medium and for any purpose provided that it is properly attributed. For attribution, the original author(s), title, publication source (PeerJ) and either DOI or URL of the article must be cited.
License URL: https://creativecommons.org/licenses/by/4.0/

Keywords: Ozone, Sensitivity, Coal chemistry, Loess Plateau, Characteristics, OBM simulation

Funding: The Fundamental Research Funds for Central Public Welfare Scientific Research Institutes of China, CRAES No. 2019YSKY-012 No. 2019YSKY-018 National Key Research and Development Program of China No. 2019YFC0214800 The National Nature Science Foundation of China No. 41907197 This work was supported by the Fundamental Research Funds for Central Public Welfare Scientific Research Institutes of China, CRAES (No. 2019YSKY-012 and 2019YSKY-018), the National Key Research and Development Program of China (No. 2019YFC0214800), and the National Nature Science Foundation of China (No. 41907197). The funders had no role in study design, data collection and analysis, decision to publish, or preparation of the manuscript.

==============================
In this study, ambient ozone (O3) pollution characteristics and sensitivity analysis were carried out in Yulin, a city in the central area of the Loess Plateau during 2017 to 2019 summer. O3 concentrations increased for 2017 to 2019. Correlation and statistics analysis indicated high temperature (T > 25 °C, low relative humidity (RH < 60%), and low wind speed (WS < 3 m/s) were favorable for O3 formation and accumulation, and the O3 pollution days (MDA8 O3 > 160 µg/m3) were predominantly observed when the wind was traveling from the easterly and southerly. O3 concentration in urban area of Yulin was higher than that in background. The pollution air masses from Fenwei Plain increase the level and duration of O3 pollution. In order to clarify the formation mechanism and source of O3, online measurements of volatile organic compounds (VOCs) were conducted from 7 July to 10 August in 2019. The average of VOCs concentration was 26 ± 12 ppbv, and large amounts of alkenes followed by aromatics, characteristic pollutants of the coal chemical industry, were detected in the ambient air. To further measure the sensitivity, the observation-based model (OBM) simulation was conducted. Empirical Kinetic Modeling Approach (EKMA) plot and relative incremental reactivity (RIR) value indicated Yulin located on VOCs-limited regime. That implied a slight decrease of NOx may increase O3 concentration. When the emission reduction ratio of anthropogenic VOCs/NOx higher than 1:1, the O3 will decrease. O3 control strategies analysis shows that the O3 targets of 5% and 10% O3 concentration reductions are achievable through precursor control, but more effort is needed to reach the 30% and 40% reduction control targets.

Introduction

China’s ozone (O3) concentrations have shown a gradual upward trend over recent years. Studies on China’s O3 pollution in China have focused on more developed regions, such as Beijing-Tianjin-Hebei (BTH) (Duan et al., 2008; Xu et al., 2011; Zhang et al., 2014b; Yan et al., 2016b), the Yangtze River Delta (YRD) (Wang et al., 2020b; Zhang et al., 2020b), the Pearl River Delta (PRD) (Chen et al., 2019; He et al., 2019), and the Guanzhong Plain (GZP) (Hui et al., 2021), which is close to the Loess Plateau. However, few studies have been conducted on the Loess Plateau itself (Li et al., 2020a; Li et al., 2020b; Wang et al., 2012). The Loess Plateau is to the north of central China, and is one of the country’s four major plateaus. Since much of its arid surface is exposed, it becomes a source of suspended particulate matter (PM) that pollutes cities in north China during dry and windy weather (Cao et al., 2008a). China began carrying out vegetation rehabilitation programs in 1978 (Li et al., 2017). With stricter air pollution control and the increase of comprehensive studies on PM between 2014 and 2017 (Cao et al., 2008a; Cao et al., 2008b; Liu et al., 2004; Wang et al., 2011; Zhang et al., 2014a; Xiaoye, 2002), the PM2.5 pollution problem plaguing Loess Plateau, and many other parts of China (Cao et al., 2012; Zhang et al., 2019), has been effectively alleviated, while O3 concentrations continue to rise. O3 has become the primary pollutant affecting Loess Plateau’s ambient air quality in the summer.

Yulin (38.25N, 109.73E) is located in the central area of Loess Plateau, in the northernmost part of Shaanxi Province (Zhu et al., 2019). It is in the junction area of Jin-Shan-Meng-Ning (Shanxi, Shaanxi, Inner Mongolian, and Ningxia provinces), which is one of China’s five main coal production areas and one of the main battlefields for coal exploitation in China (Hao et al., 2019). Yulin is a typical energy resource city, with proven coal reserves of 2.7 ×103 billion tons. In 2019, the output of raw coal alone was 4.6 ×105 kilo-tons. The area’s rapid development and energy advantages (Zhai et al., 2020) have caused many environmental problems. Previous research on this area mainly focused on PM. Lei et al. (2018) found that Yulin’s high Ca2+ levels during winter and autumn are attributed to fugitive dust, while high spring and summer sulfate levels are caused by different sources. Ion balance studies illustrated that PM10 samples were more alkaline than PM2.5 samples. Wang et al. (2011) found that dust aerosol invading Yulin could be from the northwestern desert in China and the Gobi Desert in Mongolia. CO and PM10 levels decreased while PM2.5, O3, and NO2 levels increased (Fig. S1). PM2.5 and O3 were the primary pollutants in winter and summer, respectively. Very few studies have reported on this rise of O3 concentrations (Li et al., 2020a; Hou et al., 2006). It is critical to explore the effect of local generation and transport on Yulin’s O3 levels.

In this paper, we analyzed the temporal and spatial characteristics of O3 and the influence of meteorological factors from 2017–2019. We used the observation-based (OBM) model to study the sensitivity of O3 formation to VOCs and NOx between July and August 2019. Additionally, we set several targets in our discussion of O3 control strategies.

Materials & Methods

Measurement data

To evaluate Yulin’s O3 characteristics between 2017 and 2019, we obtained observational hourly concentrations of air pollutants including O3, CO, NO2, PM10, PM2.5, and NO, as well as meteorological parameters including temperature (T), relative humidity (RH), wind speed (WS), and wind direction (WD) from National Environmental Monitoring Stations and the Yulin Ecology and Environment Bureau. In Yulin, we chose four stations for study: the Environmental Monitoring Building (EMB, 38.25°N, 109.73°E), Experimental Middle School (EMS, 38.28°N, 109.73°E), Century Square (CS, 38.29°N, 109.48°E), and Hongshixia Gorge Eco-Park (HGEP, 38.33°N, 109.74°E). The locations of the four stations are presented in Fig. 1. There are no significant local sources close to these four sites. Ambient Air Quality Monitoring Specification (on trial) (Chinese HJ 664-2013) classified the EMB site, EMS site, and CS site as urban assessing stations and the HGEP site as an urban background station. Urban assessing stations are set up to monitor ambient air quality and changing trends in urban built-up, and participate in the assessment of urban ambient air quality. Urban background stations monitor the ambient air quality in urban areas that are not affected by local urban pollution and do not participate in the assessment of urban ambient air quality. The EMB site is surrounded by a residential area and processing, manufacturing, and coal chemical industries are located about 5 km south. The EMS site and CS site are in mixed functional areas of commerce, transportation, and residences. Although the HGEP site is regulated as an urban background station, it was surrounded by a highway during our field survey and many trucks passed through it.

To investigate the characteristics of VOCs and their contribution to O3 formation in Yulin, we coupled online gas chromatography with a mass spectrometer and a flame ionization detector (GC-MS/FID, AC-GCMS 1000) developed by Guangzhou Hexin Instrument Co., Ltd. This was used to measure VOC levels at the EMB site from 7 July to 10 August 2019. A total of 70 VOCs (18 alkanes, 11 alkenes, one ethyne, 16 halocarbons, 14 aromatics, and 10 oxygenated VOCs (OVOCs)) were measured concurrently with 1 h resolution. The air passed over two cold traps in the separate FID and MS routes, and samples were rapidly heated and vaporized at a rate of up to 50 °C/s. The VOCs were brought into the FID and MS for online qualitative and quantitative analysis, respectively. The C2-C4 hydrocarbons were analyzed by the FID and the high carbon components were analyzed by the MS. To check the stability of the monitoring system, we injected the mixed standard gas (Photochemical Assessment Monitoring Stations (PAMS)) and the US Environmental Protection Agency toxic organics (TO-15)) with a concentration of 1ppb into the system seven times, and calculated the relative standard deviation (RSD) of the concentration for each component. The results showed that the RSD of all VOCs was lower than 10% (Table S1). A seven-point calibration curve was made using the standard gas external standards with the linearity (R2) of each VOC species exceeding 0.99. Another standard gas was injected daily into the instrument to test the system performance at 23:00. Therefore the data from 23:00 local time (LT) were excluded in this study.

Figure 1 Locations of four National Environmental Monitoring Stations (HGEP, CS, EMS and EMB).

(Map data ©2020 Google Earth).

Backward trajectory analysis

The 48 h backward trajectories from May to August 2017 to 2019 were computed using the Hybrid Single-Particle Lagrangian Integrated Trajectory (HYSPLIT) model (Cohen et al., 2015; Draxler & Hess, 1998) with 6-h intervals and a start time of 00:00 LT to identify the origins of air masses 1,000 m above sea level. The meteorological data were provided by the Global Data Assimilation System of the United States National Centers for Environmental Prediction (ftp://arlftp.arlhq.noaa.gov/pub/archives/gdas1). These backward trajectories from 2017, 2018, and 2019 were analyzed using HYSPLIT, and then were categorized into a few clusters for analysis of the influence of O3 regional transported sources.

Observational-based model

The OBM was developed by Cardelino & Chameides (1995) and was built on the newest version of the Master Chemical Mechanism (MCM v3.3.1; http://mcm.leeds.ac.uk/MCM/), a near-explicit mechanism describing the oxidation reactions of 146 primary VOCs together with the latest inorganic chemistry from the International Union of Pure and Applied Chemistry evaluation (He et al., 2020). Hourly concentrations of VOCs and four trace gases (SO2, CO, NO, and NO2), as well as hourly meteorological parameters (temperature and relative humidity), were used as inputs to constrain the model (Lyu et al., 2019). Moreover, this model can be employed to assess the sensitivity of O3 photochemical production by calculating the relative incremental reactivity (RIR) and changing the concentrations of its precursors without detailed or accurate knowledge of these emissions (He et al., 2019): RIRX=PO3X−PO3X−ΔX∕PO3XΔSX∕SX

where X is a precursor of O3 and △X represents the change in X concentrations; P(O3) is the O3 net production rate, which is calculated by the OBM; S(X) is the measured concentration of precursor X, and △S(X)/S(X) represents the relative change of S(X) (20% S(X) in this study).

Positive matrix factorization model

The positive matrix factorization (PMF) model is used to interpret source types and contributions based on VOC concentrations obtained from receptor point measurements (Paatero & Tapper, 1994). It has been used for the source apportionment of VOCs (Sun et al., 2020). In this study, we used PMF 5.0 to identify and quantify the main VOC sources in Yulin. A detailed description of the PMF model can be found in the EPA PMF 5.0 User Guide. Briefly, the ambient VOC observations matrix (x) was decomposed into two matrices: source profile (f) and source contribution (g): xij= ∑k=1pgikfkj+eij

where, xij is the concentration of species j measured in sample i; gik is the contribution of the source k to the sample i; fkj isthe profile of the species i in the source k; eij is the residual of species j in sample i; and p represents the total number of sources (Fan et al., 2020). In this study, we measured 70 VOC species and divided these species into three categories according to their signal-to-noise ratio (S/N) and detection limit (BDL). In general, species were classified as strong if the S/N was >2, weak if the S/N was between 0.2 and 2 or if the BDL percentage was >50%, and bad if the S/N was <0.2 or if the BDL percentage was >60% (Hui et al., 2020). Ultimately, 46 VOC species were selected and served as inputs of the PMF model.

In the PMF model, Q was the sum of the squared residual weighted by the inverse of their respective measurement uncertainty and in order to minimize the value. Q can be described as: Q= ∑i=1n ∑j=1mXij− ∑k=1pgikekjSij

Ideally, the modeled Q value should eventually approach the expected Q values (Qexp), which are equal to the degree of freedom of the model solution (n ×m –p(n+m) (Yan et al., 2016a).

Results

Temporal O3 distribution characteristics

According to the Chinese Technical regulation for ambient air quality assessment (on trial) (Chinese HJ 663-2013), 1-hour average O3 concentrations and daily maximum 8-hour average O3 concentrations (MDA8) are employed to describe hourly and daily O3 variations, respectively. The 90th percentile of daily maximum 8-h average of O3 concentrations (MDA8-90) is employed to evaluate monthly and annual O3concentration levels. In this study, we used both MDA8 and maximum daily 1-hour average O3 concentrations (MDA1) to present daily mean and extreme O3 concentrations. The Ambient Air Quality Standard (Chinese GB3095-2012) regulates the Grade II Standard of MDA8 and MDA1 O3, and are 160 µg/m3(about 75 ppbv) and 200 µg/m3 (about 93 ppbv), respectively. We defined the day when MDA8 or MDA1 O3 levels exceeded the Grade II Standard as O3 pollution day. Moreover, we also evaluated the diurnal, monthly, and annual variations observed in Yulin. The HGEP site was an urban background station which, according to technical regulation for the selection of ambient air quality monitoring stations (on trial) in China, are not included in the evaluation of a city’s air quality. Therefore, the HGEP site data were not included in the description of ambient O3 changes in Yulin.

As shown in Table 1, the mean MDA8 O3 concentration, MDA8-90 O3 concentration, and pollution days all showed a rising trend from 2017 to 2019. Previous studies found that the average MDA8 O3 concentration for China’s 31 capital cities in 2013 and 2017 was 61 and 76 ppbv, respectively (Zeng et al., 2019). In the YRD, the average MDA8 O3 concentration was 78 ppbv in 2018 (Li et al., 2019). In Lanzhou, China, the average MDA8 O3 concentration from 2013 to 2015 was 68.26 µg/m3 (about 32 ppbv) (Wang et al., 2017b). The average MDA8 O3 concentration in the GZP was 96 µg/m3 (about 45 ppbv) in 2019 (http://sthjt.shaanxi.gov.cn/zfxxgk/hjzl/hjzkgb/20201130/64307.html). Generally, O3 concentrations in Yulin were lower than developed areas such as cities in the YRD, comparable to the national average level and the GZP, and higher than Lanzhou.

The O3 pollution days all occurred between May and August in 2017 and 2018. One unanticipated finding was that O3 pollution days also appeared in April and September in 2019. The T, RH, and WS averages in April and September 2019 were 15 °C and 19 °C, 43% and 65%, and 1.7 m/s and 1.3 m/s, respectively. The T, RH, and WS averages in April and September 2017 to 2019 were 14 °C and 18 °C, 41% and 61%, and 1.8 m/s and 1.5 m/s, respectively (Table S2). The meteorological conditions in April and September 2019 were not significantly different from those in 2017 and 2018. Two O3 pollution days occurred consecutively from 18 to 19 April 2019. On 18 April, the O3 concentration peaked at 19:00 LT, decreased to 51.3 ppbv the next day at 5:00, and then started to increase again. Relatively high O3 concentration at night could be explained by regional transportation. Six consecutive O3 pollution days occurred from 3 to 8 September 2019. Unlike the O3 pollution days in April, regional transportation was not the primary reason for this O3 pollution episode. The average NO2 concentrations from 3 to 8 September were 20, 22, 27, 28, 22, and 21 ppbv, respectively, which were higher than the average concentrations on adjacent days, such as 2 September (16.6 ppbv) and 9 September (12.8 ppbv). This suggests that the continued increase in O3 concentration may have been closely related to the increase in nitrogen dioxide concentration.

Table 1 Mean daily maximum 8-h average (MDA8) O3 concentrations, the 90th percentile ofMDA8 O3 (MDA8-90) concentrations and number of pollution days from 2017–2019.

Year	MDA8-90 O3(ppbv)	Mean MDA8 O3(ppbv)	Number of pollution days	
2017	72	47 ± 18	26	
2018	72	47 ± 19	25	
2019	75	48 ± 19	34	

Figure 2A displays the monthly O3 variations. The highest MDA8-90 O3 concentrations in 2017, 2018, and 2019 were 40, 39, and 39 ppbv, respectively. The highest MDA8-90 O3 concentration appeared in May 2017 and June 2019. MDA8-90 O3 concentrations in May to July 2017-2019 exceeded the Grade II Standard. Additionally, the MDA8-90 O3 concentration in April and September 2019 also exceeded the standard. Monthly O3 variations reflected an obvious seasonal cycle, as O3 concentrations were high in summer and spring and low in winter and autumn. Some areas in north China also show this kind of seasonal cycle (Fang et al., 2020; Yan et al., 2016b; Wang et al., 2012; Li et al., 2020a). This seasonal O3 variation was largely influenced by meteorological conditions (Wang et al., 2020b; Zheng et al., 2019).

The diurnal variation of O3 concentrations is driven by photochemical reactions, meteorological conditions, the transport of O3 and its precursors, and surface deposition (Crutzen, 2016; David & Nair, 2011). The diurnal variation in Yulin from 2017-2019, similar to many cities in northern China, shows a typical single peak curve: a valley in early morning and a peak in afternoon (Fig. 2B; Wang et al., 2012; Yu et al., 2020b; Li et al., 2020a). It was apparent that the O3 concentrations were highest in summer, then spring, autumn, and winter. The O3 concentration peaks were 60, 67, 43, and 34 ppbv in spring, summer, autumn, and winter, respectively. The peaks occurred at 16:00 LT in spring, autumn, and winter, but at 15:00 LT in summer. The solar radiation on Loess Plateau is intense, especially in summer, which can lead to more efficient O3 production (Hu et al., 2014).

Figure 2 (A) Monthly variation of MDA8-90 O3 concentrations, (B) diurnal variation of 1-hour average O3 concentrations.

Spatial O3 distribution characteristics

The four Yulin sites from most south to most north are the EMB, EMS, CS, and HGEP. Figure 3 shows that O3 concentrations decreased spatially from south to north between 2017 and 2018. In 2017, the EMB site had the highest O3 concentration, followed by the CS site, then the EMS site, and the HGEP site. In 2018, the O3 concentrations decreased from south to north. In 2019, the EMB site had the highest O3 concentration, followed by the HGEP site, then the EMS site, and the CS site, implying that the central region had lower O3 concentrations than the northern and southern regions. The EMB site had the highest concentrations between 2017 and 2019. This suggested that Yulin’s O3 concentration was greatly affected by industry. Diurnal and monthly variations at the four sites showed the same trend (Fig. S2). The diurnal variation curve showed that the lowest concentrations appeared at 7:00 LT in the morning and the highest concentrations appeared at 16:00 LT in the afternoon. The distance between these sites is less than 5 km. Therefore, different pollution sources can explain the differences in O3 concentrations at the four sites.

Figure 3 O3 concentrations distributed at four sites in Yulin from 2017 to 2019.

In each box plot, the block is the mean, the horizontal line crossing the box is the median, the bottom and top of the box are lower and upper quartiles, and the whiskers are the 10th percentile and 90th percentile.

The HGEP site’s O3 concentration increased from 2017 to 2019 but was lower than that of Yulin’s urban area. The monthly and daily changes were consistent with those of the urban area. However, the O3 concentrations were lower at night at the HGEP site than at the other sites (Fig. S2A). As mentioned above, the HGEP site is a traffic-intensive area, and vehicles emit NO that can titrate and lower O3 levels.

Overview of O3 pollution during the VOC sampling period

To further reveal the impact of O3local formation, we collected VOC samples at the EMB site from 7 July to 10 August 2019. Figure 4 shows the time series of the meteorological factors and NO2, CO, PM2.5, PM10, VOC, and O3 concentrations in Yulin during the VOC sampling period. The average values are listed in Table S3. We regarded consecutive MDA8 O3 pollution days as one O3 episode period (EP). There were a total of six O3 episode periods (EP1: 11-14 July, EP2: 20 July, EP3: 25-26 July, EP4: 28 July, EP5: 30 July-2 August, and EP6: 10 August) observed during the sampling period. During the sampling period, the average MDA8 O3 concentration was 72 ± 9 ppbv with a range from 50 ppbv to 85 ppbv, and 83 ± 2, 75, 82 ± 3, 77, 79 ± 3 and 84 ppbv, respectively, from EP1 to EP6. The average concentration of total VOCs was 26 ± 12 ppbv with a wide range from 6 to 90 ppbv, and 30, 41, 28, 27, 31, and 13 ppbv, respectively, in the EPs. The meteorological factors of these six EPs (Table S3) were beneficial to in situ photochemical processes. NO2 and total volatile organic compound (TVOC) concentrations were higher than in the non-pollution period. This indicated an increase in local precursor emissions. The valley values on 26 July (EP3), 28 July (EP4), 2 August (EP5), and 10 August (EP6) were 31, 22 50, and 33 ppbv, respectively. Compared to the average valley value during the VOC sampling period (19 ppbv) and the median value (16 ppbv), these valley values were relatively high. This result implied that EP3, EP4, EP5, and EP6 were also affected by regional transportation. The 48 h backward trajectories when hourly O3 concentrations were over 75 ppbv are shown in Fig. S3. Most of the backward trajectories of EP1 and EP2 came from local and southwest areas, respectively, and there were no O3 pollution areas along the trajectories. Most of the backward trajectories of EP3, EP4, and EP5 came from the Fenwei Plain, and O3 pollution also occurred in the cities along the trajectories (Table S4). However, during EP6, no pollution was found along the trajectories. This O3 EP was complex and the data cannot be explained. Determining whether the changes in O3 concentration were caused by transportation, meteorological conditions, or anthropogenic sources will be our focus in future studies.

Figure 4 Time series of WS, WD, T, RH, O3, PM2.5, PM10, NO2, CO and VOCs at the EMB site during VOCs sampling period.

Different VOC species are capable of influencing O3 formation potential (OFP), which can be estimated using Maximum Incremental Reactivity (MIR) (Carter, 2009). The OH reactivity concept is useful when estimating VOC O3 production because VOC degradation caused by OH oxidation eventually leads to net O3 production (Tan et al., 2018). OH reactivity is the sum of the products of precursor O3 concentrations and the reaction rate constants between O3 precursors and OH (Atkinson & Arey, 2004). The percentage contributions of alkanes, alkenes, ethyne, halocarbons, aromatics, oxygenated VOCs, and isoprene to the mixing ratio, LOH, and OFP of total VOCs are shown in Fig. 5. Among these, alkanes were the most abundant VOC species, accounting for 46.2%, followed by OVOCs (26.9%), alkenes (10.1%), aromatics (7.8%), ethyne (4.3%), halocarbons (4.3%), and isoprene (0.4%). Although alkanes were the most abundant VOC species in Yulin, the proportions of LOH (12.7%) and OFP (14.0%) were relatively small. Alkenes (31.8%), OVOCs (31.7%), and aromatics (21.3%) were the top three VOC species in terms of OFP proportion. LOH and OFP showed good consistency, and alkenes (44%), OVOCs (23.5%), and aromatics (12.8%) had the top three proportions of LOH. In previous studies, aromatics accounted for the highest OFP and LOH proportions in the PRD (Yu et al., 2020a; Tan et al., 2019). In Xi’an, the capital of Shaanxi Province, alkenes have been shown to have the highest proportion of LOH and OFP (Song et al., 2020). The OFP of sampling sites around Lanzhou were mainly from aromatics, which accounted for 46.3% (Wu et al., 2019), due to the petrochemical industry there. Due to its abundant coal resources, Yulin has many coking plants. The exhaust gas from coking plants contains a lot of alkenes, the key species in O3 formation, and aromatics (Zhang et al., 2020c).

Figure 5 The ratio of concentrations, OH reaction (LOH) and O3 formation potential (OFP) of alkanes, alkenes, ethyne, halocarbons, aromatics, oxygenated VOCs (OVOCs) and isoprene.

Discussion

The impact of meteorological parameters on O3 concentration

Liu & Wang (2020) used the Community Multiscale Air Quality (CMAQ) modeling system and found that the impact of meteorological factors on O3 trends varied by region and by year. Between 2013 and 2017, O3 concentrations in China were affected by meteorological factors, which showed comparable or even greater effects than anthropogenic emissions. Meteorological conditions such as T, RH, WS, and WR directly affect surface O3 concentrations via changes in chemical reaction rates, dilution, wet and dry removal, and transport flux, or indirectly via changes in natural emissions (Lu, Zhang & Shen, 2019b; Li et al., 2020c). We focused on the relationship between meteorological parameters (T, RH, WS, and WR) and O3 concentrations between May to August from 2017 to 2019 because O3 pollution days mainly occurred during this period. The correlational analysis results are shown in Table 2. Table S5 shows the total number of days, number of MDA8 and MDA1 O3 pollution days, and MDA8 and MDA1 O3 exceedance probabilities in different ranges of T, RH, WS, and WR. The O3 exceedance probability was calculated using the percentage of the number of O3 pollution days across the total number of days. The influence of wind speed on ground O3 concentrations was insignificant (−0.088). This result matches those in earlier studies (Chen et al., 2020; Fan et al., 2020).

Table 2 Relationship between daily average temperature (T), relative humidity (RH) and wind speed (WS) and MDA8 and maximum daily 1-hour average O3 concentrations (MDA1).

	T	RH	WS	
MDA8	0.547**	−0.286**	−0.088**	
MDA1	0.519**	−0.250**	−0.164**	
Notes.

** Correlation is significant at the 0.01 level.

T directly impacts chemical kinetic rates and the mechanistic pathway for O3 formation (Atkinson, 1990). High RH days were always associated with more cloud cover, which can reduce photochemistry (Kavassalis & Murphy, 2017). The correlation coefficients between T and MDA8 O3 and MDA1 O3 were 0.547 and 0.519, respectively. Between RH and MDA8 O3 and MDA1 O3, the correlation coefficients were −0.286 and −0.250, respectively (Table 2). O3 was positively correlated with T and negatively correlated with RH. Further statistical test results are shown in Table S5. The MDA8 (MDA1) O3 exceedance probability of T in the ranges of 10−20 °C, 20−30 °C, and over 30 °C were 0.8% (0%), 27.7% (6.6%), and 81.82% (9.1%), respectively, and that of RH in the ranges of 20–40%, 40–60%, 60–80%, and 80–100% were 22.6% (6.1%), 31.3% (9.0%), 19.3% (2.3%), and 1.04% (0.0%), respectively. These results confirm that there is a correlation between O3 and T and RH. When T was higher than 30 °C and RH was in the range of 40–60%, the probability of O3 pollution events was higher.

The correlation coefficients between WS and MDA8 O3 and MDA1 O3were −0.088 and −0.164, respectively, which indicated that the influence of WS on ground O3 concentrations was insignificant. Table S5 shows that the MDA8 (MDA1) O3 exceedance probability of WS in the range of 0–1.5 m/s, 1.5–3 m/s, 3–4.5 m/s, and greater than 4.5% were 22.5% (5.8%), 20.36% (3.3%), 9.26% (1.85%), and 0% (0%), respectively. As WS increased, the exceedance probability showed a downward trend. Generally, the increase in WS caused O3 concentrations to decrease due to enhanced dry deposition, mixing, and dilution (Dawson, Adams & Pandis, 2007). Additionally, WS and WR affected O3 concentrations by transporting O3 and its precursors. The MDA8 (MDA1) O3 exceedance probability of northerly, easterly, southerly, and westerly winds were 11.2% (5.1%), 28.0% (3.7%), 26.4% (5.7%), and 17.2% (5.7%), respectively. The effects of T, RH, and WS on MDA8 O3 and MDA1 O3 were similar. In contrast, the easterly winds showed the largest O3 exceedance probability of MDA8 O3 and the southerly and westerly winds showed the greatest MDA1 O3 exceedance probability. The results implied that T and RH can directly affect O3 formation and loss. However, the influence of wind on O3 concentrations was complex. In general, high T, low RH, and low WS were shown to cause O3 pollution, especially when the T exceeded 25 °C, the RH was less than 60%, and the wind speed was less than 3 m/s. O3 pollution days were predominantly observed when the wind was traveling from the east and south.

Regional transportation

Besides local formation, long-distance regional transportation also plays an important role in increasing O3 concentration (Hui et al., 2018). Figures 6A–6C show the clustering analysis for May to August 2017 to 2019. In this study, when O3 concentrations were greater than 75 ppbv, we identified this backward trajectory as a polluted trajectory. In 2017, a total of 487 backward trajectories were categorized into five clusters. Cluster a1 to a5 accounted for 49%, 21%, 3%, 13%. and 10% of the total trajectories, respectively. There was a total of 44 polluted trajectories in 2017. Cluster a1 to a5 contained 27, five, two, one, and eight polluted trajectories, respectively. Cluster a1’ s air masses from Weinan in Shaanxi Province that crossed Yan’an to Yulin had the greatest impact on O3 concentrations in Yulin in 2017. In 2018, there was a total of 492 backward trajectories across five clusters. Cluster b1 to b5 accounted for 13%, 23%, 27%, 31%, and 5% of the total trajectories, respectively. There was a total of 40 pollution trajectories in 2018. One, 10, 13, 12, and four pollution trajectories were grouped into clusters b1 to b5, respectively. Therefore, when the results of the cluster and the distribution of pollution trajectories were combined, the air masses of clusters b2, b3, and b4 had the greatest impact on the O3 in Yulin in 2018. Cluster b2 was from Inner Mongolia. Cluster b3 came from Yan’an to Yulin. Cluster b4 came from Kaifeng in Henan Province, passing through Zhengzhou, Luoyang, Yuncheng in Shanxi Province, and Yan’an in Shaanxi Province. In 2019, a total of 492 backward trajectories were divided into three clusters. Cluster c1 to c3 accounted for 16%, 33%, and 51% of total trajectories, respectively. There was a total of 42 pollution trajectories in 2019. Thirty-two and 10 pollution trajectories were grouped into cluster c2 and c3, respectively. Therefore, the air masses of cluster c2, which came from Linfen in Shanxi and passed through Weinan and Yan’an in Shaanxi Province to Yulin, had the greatest impact on the O3 in Yulin in 2019. Fenwei Plain is to the southeast of Yulin and has 11 cities including Weinan, Luoyang, and Yuncheng, mentioned above. Fenwei Plain has a high O3 concentration because it has more industry and a larger population (Lu et al., 2019a; Wang et al., 2020a). The airflows from Yan’an and FenWei Plain between May and August may aggravate O3 pollution in Yulin.

Figure 6 Result of cluster analysis of backward trajectories in Yulin from May to August 2017 (A), 2018 (B), 2019 (C).

And the number of polluted trajectories in every cluster. The backward trajectory with O3 concentrations over 75 ppbv is identified as a polluted trajectory.

Sensitivity analysis

The relationships between O3 and VOCs and NOx are nonlinear. In order to establish effective O3 control measures, these relationships must be explored. In this study, we applied two methods to evaluate O3-VOC-NOx sensitivity in Yulin: the RIR, which was mentioned in sec2.3, and the EKMA (Carter, Winer & PittsJr, 1982). The average hourly VOC concentrations during the sampling period were put in the OBM model as a base scenario. By increasing or decreasing the ratio of precursors, we were able to simulate the changes in MDA8 O3. The relationship between MDA8 O3 and the relative changes of anthropogenic VOCs (AVOCs) and NOx can be expressed using a contour plot for MDA8 O3, which we named EKMA (Fig. 7A). Isoprene was mainly emitted by the biogenic source and was difficult to control. Therefore, the AVOCs excluded isoprene. The EKMA plot was split into two parts by a ridgeline that denoted the local maxima of the rate of O3 formation (Li et al., 2020b; Hui et al., 2018). The upper-left and lower-right areas represent the O3 formation under VOC-limited and NOx-limited conditions, respectively. In Fig. 7A, the circles and pentagram represent pollution days and the base scenario, respectively. These points were all located in the VOC-limited regime and showed the negative effects of NOx reduction in response to O3 production control. Previous studies reported that in China, unban areas, rural regions, and urban agglomerations were under VOC-limited, NOx-limited, and VOC-NOx-limited conditions (Liu et al., 2019; Wang et al., 2017a), respectively.

Figure 7 EKMA and Relative incremental reactivity (RIR).

(A) Contour plot of MDA8 as a function of VOC reactivity and NOx concentrations (EKMA), the star represents the base scenario and circles represent O3 pollution days (A). (B) Relative incremental reactivity (RIR) for AVOCs, isoprene, CO and NOx on O3 pollution days.

Figure 7B shows the RIR values for AVOCs, isoprene, CO, and NOx during pollution days. The RIR values for AVOCs were in the range of 0.73–2.83%/%, which was significantly higher than those for isoprene (0.22–0.42%/%) and CO (under 0.01%/%). This suggested that AVOCs played a more significant role in O3 formation during these EPs, while the influence of isoprene, especially CO, was negligible. Excluding 26 July, the RIR values for NOx were between −0.26 and −0.10%/%, indicating that the O3 formation during these days was in the VOC-limited regime. This result was consistent with that of EKMA. However, on 26 July, the RIR values of NOx and VOCs were 0.60 and 0.73, respectively, and these two positive values with a small difference implied that the O3 formation on 26 July was controlled by both VOCs and NOx. This result was contrary to that of the EKMA plot, which may be because the EKMA simulation was based on the average of the sampling period. The average RIR values for AVOCs from EP1 to EP6 were 1.79%/%, 1.92%/%, 1.11%/%, 1.13%/%, 1.34%/%, and 2.83%/%, respectively, and for NOx were -0.99%/%, 1.26%/%, -0.14%/%, -0.10%/%, -0.68%/%, and -2.36%/%, respectively. This demonstrated that these EPs were all in the VOC-limited regime. The absolute RIR values of the AVOCs were larger than those of the NOx, suggesting that O3 formation was most sensitive to AVOC reduction.

O3 control strategies

It is necessary to discuss how much AVOCs should be controlled for the most efficient O3 reduction. Using the sampling period as an example, we will discuss O3 control measures. Figure 8A shows the increments of the MDA8 O3 response (positive and negative values represented the increase and decrease in O3 compared to the base case with no VOC or NOx reductions, respectively) to different emission reductions in AVOCs and NOx (AVOC reduction/NOx reduction=1:2, 1:1, 2:1, 3:1, or 4:1, and only reducing AVOCs or NOx). The horizontal axis represents the combined reduction percentage of AVOCs and NOx. For instance, a total reduction percentage of 120% indicates that there were both 60% reductions in AVOCs and NOx emissions for AVOC/NOx=1:1, or 80% and 40% reductions in the AVOC and NOx emissions, respectively, for AVOC/NOx=2:1. When the AVOC/NOx ratio was more than 1:1, namely if the AVOC emission was reduced more than NOx, the O3 concentration would decline. When a certain O3 reduction target was achieved, the VOC and NOx ratio was higher and the total emission reduction was lower. For example, to reduce O3 concentration by 5%, the total reduction emissions of AVOC and NOx were 12%, 20%, and 174% for an abatement AVOC/NOx ratio of 4:1, 2:1, and 1:1, respectively, or by only cutting AVOC emissions by 7%. Only cutting AVOC emissions seemed to be the most efficient way to control O3. However, NO2 is one of the most significant ambient air pollutants that can influence ambient air quality. NOx needs to be reduced since it is an important precursor of PM2.5. It is important to cooperatively control the emission of AVOCs and NOx. During sampling periods, the mean and highest MDA8 O3 concentrations were 79 and 102 ppbv, respectively, which exceeded the Nation Grade by 5% and 36%. According to this, we set four O3 control targets, 5%, 10%, 30%, and 40%, to explore the abatement percentages of AVOCs and NOx (Fig. 8B). To achieve the O3 control target of 5%, AVOC emissions needed to be reduced 10%–90%, with NOx also in the range of 10%–90%. To achieve the O3 control target of 10%, AVOC emissions needed to be reduced 20% to 90% with NOx in the range of 5% to 80%. However, to achieve the targets of 30% and 40%, the AVOC emissions needed to be reduced by at least 60% and 80%, respectively. Even though more stringent VOC and NOx control measures have been implemented, it is still challenging to achieve the 30% and 40% O3 control objectives.

Figure 8 (A) The increment percentage of MDA8 O3 are shown under different NOx and VOCs reduction pathways and (B) the reduction percentages of NOx and VOCs.

Qtrue/Qexpected was an appropriate index to decide how many factors were optimal in the PMF model. Here, we evaluated four to 10 factor solutions and found that the Qtrue/Qexpected values of different resolved numbers and that eight to nine factors were the lowest (Fig. S2). Therefore, we identified eight types of major pollution sources using PMF source analysis (Fig. 9 and Fig. S3). Factor 1 had a high percentage of aromatics and a relatively high ratio of toluene. In previous studies, toluene was the most abundant VOC species in painting and factor 1 was considered to be the source of the solvent (Wang et al., 2014). Coking plants and the petrochemical industry are pillars in Yulin and emit a lot of ethene (Zhang et al., 2020c; Han et al., 2018), so we considered factor 2 to be industrial sources. The main source of isoprene in the city is plant emissions (Zhang et al., 2020a; Loivamaki et al., 2007) and factor 3 is considered a biological source. Factor 4 was characterized by a high percentage of ethane and high loadings of OVOCs. Previous studies in Shenzhen and Wangdu found that background pollution largely contributed to OVOCs (Han et al., 2019). Factor 4 was considered background pollution. High percentages of propane, acetylene, n-butane, cyclohexane, ethylbenzene, o-xylene, isobutane, m/p-xylene, and n-hexane are found in vehicle exhaust (Fan et al., 2020), and these characteristics are consistent with factors 5 and 7. The proportion of isopentane in gasoline exhaust is more than that found in diesel exhaust (Ly et al., 2020). The proportion of aromatic hydrocarbons is higher in gasoline vehicles than in diesel vehicles because aromatic hydrocarbons are the main contributor to octane in gasoline, and gasoline is more volatile than diesel (Huang et al., 2020). Factors 5 and 7 were considered gasoline exhaust and diesel exhaust, respectively. Isopentane is mainly distributed in factor 6 and is a typical marker of fuel evaporation (Liu et al., 2008), so factor 6 was considered fuel evaporation. Factor 8 was characterized by a high percentage of ethane, ethene, and 1-butene. Previous studies reported that VOCs emitted from coal combustion (Zhang et al., 2020a) are mainly C2-C3 alkenes and C2-C3 alkanes. Factor 7 was considered coal combustion. Vehicle emissions count for 51.1% of emissions (Fig. 9) and diesel exhaust is most dominant. Industrial sources account for 17.0%. Previous emission reduction strategies focused on reducing vehicle and industry emissions, especially diesel exhaust.

Figure 9 Relative contributions of different sources to VOCs during study period.

Conclusion

We measured ambient O3 levels at four National Environmental Monitoring Stations. The MDA8 90th O3 concentrations in Yulin were 72, 72, and 75 ppbv from 2017 to 2019, respectively, and they showed an increasing trend. The EMB site had the highest O3 levels. Because the HPGP site is a traffic-intensive area, this site had more NO emissions from trucks during night transportation that led to lower night concentrations than at the other sites. O3 concentrations were higher in summer and lower in winter, reflecting an obvious seasonal cycle, and the pollution days occurred between May and August.

Our O3 and T, RH, and WS correlation analysis results implied that O3 had a positive correlation with T and a negative correlation with RH. Through statistical analysis, we found that when T was greater than 25 °C, RH was lower than 60%, WS was less than 3m/s, and the WD was northerly, the probability of MDA8 O3 exceeding the Grade II Standard increased. Using the HYSPLIT model to calculate the backward trajectories from May to August 2017 to 2019, we saw that the O3 concentration in summer was greatly affected by transportation on the Fenwei Plain.

The VOC sampling period from July 7th to August 10th had 13 O3 pollution days across six EPs. During this period, the average O3 and TVOCs were 50 ± 22 ppbv and 26 ± 12 ppbv, respectively. Alkane was the most abundant VOC group. As a result of the OFP and OH reaction, alkenes were the most important VOC species for O3 formation. Using the OBM-calculated EKMA plot and RIR value indicated that Yulin is located in a VOC-limited regime. This suggested that a slight decrease of NOx may increase O3 concentration. The AVOC RIR value was the highest and was the key VOC that needed to be primarily controlled. In regard to O3 control strategies, when the reduction of AVOC/NOx is higher than 1:1, the O3 will decrease. According to the average and maximum MDA8 O3, we set four targets for decreased O3 emissions. For O3 concentrations to decrease by 5%, AVOCs and NOx are needed to reduce emissions by 10%–90%. For O3 concentrations to decrease by 10%, AVOCs and NOx are needed to reduce emissions by 20%–90% and 5%–80%, respectively. However, the 30% and 40% targets will be difficult to achieve. The PMF model showed that vehicle exhaust and coal chemical industry emissions are the main sources of AVOC emission reductions.

Supplemental Information

Supplemental Information 1 GC-MS/FID method for target compounds, Over-standard rate of ozone concentration in different ranges, average trace gas concentrations and meteorological parameters, and PMF source profile

Table S1 Summary of GC-MS/FID method for target compounds and their concentrations, Table S2 Over-standard rate of ozone concentration in different ranges of temperature, relative humidity, wind speed and wind direction, Table S3 Average trace gas concentrations and meteorological parameters during six O3 episode periods and VOCs sampling period, Fig. S1 (A) Monthly variation of MDA8-90th O3 concentrations and (B) diurnal variation of 1-hour average O3 concentrations for four sites, Fig. S2 48 h backward trajectory on 26 July (A), 1 (B) and 10 (C) August, Fig. S3 PMF source profile.

Click here for additional data file.

Supplemental Information 2 Meteorological factors, O3 and VOCs concentrations in the research area from 2017 to 2019

Click here for additional data file.

We sincerely thank Zhen He from Guizhou University, and Fang Bi, Zhenhai Wu, and Yujie Zhang from the Chinese Research Academy of Environmental Sciences for their great help. We would also like to thank the reviewers and editors who contributed valuable comments to improve the quality of this paper.

Additional Information and Declarations

Competing Interests

Author Contributions

Data Availability

The authors declare there are no competing interests.

Manfei Yin and Xin Zhang performed the experiments, analyzed the data, prepared figures and/or tables, and approved the final draft.

Yunfeng Li performed the experiments, prepared figures and/or tables, and approved the final draft.

Kai Fan analyzed the data, prepared figures and/or tables, and approved the final draft.

Hong Li performed the experiments, authored or reviewed drafts of the paper, and approved the final draft.

Rui Gao and Jinjuan Li conceived and designed the experiments, authored or reviewed drafts of the paper, and approved the final draft.

The following information was supplied regarding data availability:

The raw measurements are available in the Supplemental Files.

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
