# Peer review of "Ambient ozone pollution at a coal chemical industry city in the border of Loess Plateau and Mu Us Desert: characteristics, sensitivity analysis and control strategies"

_PeerJ, doi:10.7717/peerj.11322_

## Round 0.1 · original submission · Major Revisions

Bsed on the comments from three anonymous reviewers, your manuscpt requires some major revisions. Please revise your manuscript carefully point by point and submit the revised manuscript within one month.

Reviewer 1 ·

Basic reporting

Most of the results presented in the manuscript are lack of evidence. In addition, some description in the manuscript is unclear and confused. The following presented some examples:
1. Lines 187: The value of wind speed has been illustrated before.
2. Lines 172-173: Why the temporal variations of O3 were not discussed for HGEP site (the urban background site)?
3. Lines 187-188: does it mean that the meteorological conditions in April and September 2019 were not significantly different from those in 2017 and 2018?
4. Lines 1910197: Much more evidence was needed to demonstrate that the sustained increase in O3 from 3-8 September 2019 was mostly related to local emissions.
5. Lines 236-237: As mentioned above HGEP is an urban background site not a traffic-intensive area.
6. Line 253: What are the units for the valley values?
7. Line 254: What are the valley values in past?
8. Line 253-259: To highlight the difference between regional transport and local emissions, it is suggested to compare backward trajectories between EP3, EP4, EP6 and EP1, EP2, EP5.
9. It is suggested to provide a figure to present the time series of air pollutants from 2017 to 2019.
10. It is suggested to provide a figure to provide the statistics of VOC species, NOx, CO and NOx in different years.
11. It is suggested to provide a detailed description for the confirmation of PMF solution.
12. It is suggested to highlight the difference between local emissions and regional transport instead of just simply providing the classification of backward trajectories.
13. The results in the manuscript are lack of logic.

Experimental design

no comment.

Validity of the findings

As the results of this study were not presented properly, the findings of this study could not provide much useful information for the O3 research community and air quality management departments. Furthermore, the science values of this study needs to be highlighted.

·

Basic reporting

Professional articl strucle, figures and tables.
Sufficient field background.

Experimental design

Research question well defined.
Rigorous investigation performed to high technical and ethical standard.

Validity of the findings

All underlying data have been provided.
conclusions are well stated.

Reviewer 3 ·

Basic reporting

no comment

Experimental design

no comment

Validity of the findings

no comment

Additional comments

The manuscript investigated ambient O3 pollution characteristics and sensitivity analysis over Yulin, a typical coal-chemical industry city in China. The manuscript highlight O3 concentration reductions are achievable through precursor control, but more effort is needed to reach the 30% and 40% reduction control targets. Overall, the methodology and results seem well presented. I recommend the manuscript be considered for publication only after my following comments being addressed.
Abstract:
“OBM, EKMA, RIR” may make the readers confused. Pls give the full names.
Introduction:
Line 63-68: As the author just mentioned the O3 pollution over BTH and YRD, what about the O3 pollution over the Guanzhong Plain? Can the author make a review about the O3 pollution across China, not just BTH and YRD?
Line 79-87: For this paragraph, the author should make a review about the current atmospheric pollution studies over Yulin, for example, PM pollution?
Results:
Line 222: What is the objective of this spatial distribution of O3 pollution? More discussion is needed.
Line 357-358: Pls re-write this sentence.
In addition, I don’t quite understand the sensitivity analysis? What do you mean by sensitivity analysis of O3 pollution?
Table:
Add SD.

---

## Round 0.2 · accepted · Accept

Based on the comments from the reviewer, the manuscript can be accepted for publication on PeerJ. Thank you for your contribution to this journal. Looking forward to your next excellent work.

·

Basic reporting

This manuscript has clear and unambiguous, professional English used throught.

Experimental design

The Experimental desigh has sufficient detail and information to replicate.

Validity of the findings

The conclusions are well stated, linked to original research question.